# Height-renderable morphable tactile display enabled by programmable modulation of local stiffness in photothermally active polymer

Inwook Hwang ®[1], Seongcheol Mun[1], Jung-Hwan Youn[1], Hyeong Jun Kim[2], Seung Koo Park[3], Meejeong Choi[1], Tae June Kang[2], Qibing Pei ®[4] & Sungryul Yun ®[1]✉

Reconfigurable tactile displays are being used to provide refreshable Braille information; however, the delivered information is currently limited to an alternative of Braille because of difficulties in controlling the deformation height. Herein, we present a photothermally activated polymer-bilayer-based morphable tactile display that can programmably generate tangible three-dimensional topologies with varying textures on a thin film surface. The morphable tactile display was composed of a heterogeneous polymer structure that integrated a stiffness-tunable polymer into a light-absorbing elastomer, near-infra-red light-emitting diode (NIR-LED) array, and small pneumatic chamber. Topological expression was enabled by producing localized out-of-plane deformation that was reversible, height-adjustable, and latchable in response to light-triggered stiffness modulation at each target area under switching of stationary pneumatic pressure. Notably, the tactile display could express a spatial softness map of the latched topology upon re-exposing the target areas to modulated light from the NIR-LED array. We expect the developed tactile display to open a pathway for generating high-dimensional tactile information on electronic devices and enable realistic interaction in augmented and virtual environments.

Tactile displays have been developed to convey textual information to visually impaired people and enhance user experience via multi-modal interaction. In particular, refreshable Braille displays are being employed to replace paper Braille books, which are difficult to produce, and to convey text information on smart devices[1]. Height-adjustable tactile displays have been introduced in the form of tactile pin arrays that can express 2.5D images for tangible interaction[2–5]. However, these pin-type displays comprise a separate mechanical driver in addition to the actual moving part, thus requiring bulky hardware. Furthermore, the separate structure of the frame and pins leads to bumps and gaps between the parts, which are noticeable by touch, are prone to contamination, and cause the pins to wobble. Constructing a rigid mechanical structure that is sufficiently flexible for embedding or bending to fit into a wearable device is difficult.

[1]Tangible Interface Creative Research Section, Electronics and Telecommunications Research Institute, Daejeon, South Korea. [2]Department of Mechanical Engineering, Inha University, Incheon, South Korea. [3]Human Enhancement & Assistive Technology Research Section, Electronics and Telecommunications Research Institute, Daejeon, South Korea. [4]Department of Materials Science and Engineering, Henry Samueli School of Engineering and Applied Science, University of California, Los Angeles, CA, USA. ✉e-mail: sungryul@etri.re.kr

Recently, flexible actuator-based tactile displays have been actively investigated as alternatives. The first approach involved exploiting thermally induced stiffness changes in a thin functional polymer, which allowed the material to exist in two states: rigid or rubbery. Specifically, a polymer membrane, which is softened in a heating oven or through localized Joule heating and integrated into a perforated pneumatic pressure chamber, deformed protrusively in response to an electric field applied across the membrane. The deformed shape could then be latched by recovering its original rigidity while cooling to room temperature[6]. By electrical actuation of such a bistable polymer, Peng et al.[7] developed a tactile display that could express Braille or alphabets. By establishing stretchable micro-heaters on a bistable polymer, Besse et al.[8], realized a flexible tactile display with a 32 × 24 array that could express symbolic patterns under pneumatic pressure. enabling zero power consumption at rest and a high support force of 1 N. Meanwhile, contrary to the aforementioned approach, hydraulically amplified taxel (HAXEL), which exploits electrostatic actuation, was developed as a wearable 5 × 5 tactile array[9]. With a thin-layer polymer structure that could form a liquid-filled cavity, HAXEL exhibited an excellent response time of several milliseconds to produce an out-of-plane deformation as high as 500 μm; however, it required electrical energy to maintain the deformed state with a low support force of ~300 mN. Artificial skin was developed by utilizing the volume phase transition of hydrogels in response to a temperature field generated by light illumination[10]. The artificial skin could express the curvilinear contour of a physical shape with an intagliated depth of 250 μm via the thermal shrinking of 65 × 65 cuboid microstructures on the hydrogel layer.

A possible key to improving the availability of tactile arrays is the application of an appropriate latching strategy[1], as each tactile cell needs to support the contact force of the user's finger, up to 1 N, during normal usage[11]. To impart the latching capability to tactile cells, researchers have employed different semi-latching strategies, including the application of electric heating[7,8], magnetic force[12,13], and electromagnetic force combined with a mechanical fully latching structure[14]. These semi-latching strategies also reduce power consumption while a user scans the rendered tactile image.

In parallel, the tactile displays studied to date can produce height changes of several hundred micrometers. Most of them operate in discrete two-stage on/off modes with no intermediate height steps due to the limitations for precise control[1]. However, for haptically conveying visual images such as photographs, drawings, graphs, and maps to visually impaired people and for high-dimensional information about objects, a tactile display capable of varying levels of height expression is required. These requirements are not fully satisfied by existing tactile displays.

To this end, we developed a light-triggered morphable tactile display that can display a height map on a functional polymer bilayer via the out-of-plane deformation of individual target areas in response to photothermal heating under low and stationary pneumatic pressure, as well as lock the height map without any energy supply. The functional polymer bilayer was fabricated by forming a graphene nanoplatelet (GNP)-polydimethylsiloxane (PDMS) composite elastomer (GNPE) on a photo-crosslinked poly(*tert*-butyl acrylate) (P*t*BA) layer. The GNPE layer allowed selective regions on the P*t*BA layer to be photothermally converted to a rubbery state, enabling multilevel softness control of the bilayer with light intensity. Moreover, it complemented the insufficient mechanical resilience of the P*t*BA layer, particularly at temperatures near the $T_g$ of P*t*BA (40–70 °C). The light-induced heating strategy, which eliminated the need to attach an electrode to the polymer surface, contributed significantly to ensuring mechanical robustness against large and repeated deformations. Owing to these benefits, our tactile display generated refreshable, height-adjustable, and latchable out-of-plane deformations at each area independently, enabling precise fourteen-step control of its amplitude within a height range of 0–1420 μm with a small deviation (<5%) under most conditions. We demonstrated the ability of our tactile display to programmably morph from a flat surface into various 3D geometries and express softness and warmth on the morphed structure via photothermally induced stiffness modulation at localized regions of the polymer bilayer.

## Results

### Design concept of the morphable tactile display

Current tactile displays are limited in terms of haptically conveying visual images with a high degree of freedom due to the numerous technical challenges of imparting a versatile capability to produce rapid, refreshable, and load-bearing deformation, particularly with precise height tunability in localized areas, while ensuring durability and multimodality that provides diverse tactile sensations. To realize a tactile display with unprecedented capability, we designed a functional polymer bilayer suitable for adopting a light-driven strategy by combining a stiffness-tunable polymer and light-absorbing elastomer. Each polymer for the bilayer design had a desired core functionality. The light-absorbing elastomer was required to have low thermal conductivity to enable photothermal heating only in the light-irradiated area while maintaining sufficient softness and elasticity to prevent a reduction in the achievable deformation. In parallel, as illustrated in Fig. 1a, the stiffness-tunable polymer was required to soften progressively with increasing temperature to ensure reproducible and clear changes in modulus with varying temperatures. The use of a bilayer comprising functional polymers with these characteristics allows the tactile display to photothermally draw a stiffness map and convert it into protrusive deformation that is adjustable in inverse proportion to stiffness, as well as latchable and restorable via a stepwise process: locally changing the elastic modulus of the flat bilayer surface at a desired region by controlling the light intensity (Step I), morphing the region into a curved shape with different heights by applying stationary pneumatic pressure (Step II), latching the deformed state by cooling the bilayer as the light is turned off (step III), and recovering to the initial state by reapplying the light at atmospheric pressure (step IV). The co-stimulus-driven morphing process enabled the bilayer to not only reversibly express three-dimensional structures, physically reconfigurable user interfaces (UI), Braille, and letters (Fig. 1b) but also provide texture with warmth on the morphed shape via control of the softness and photothermally induced temperature at localized areas (Fig. 1c).

### Materials for the stiffness-tunable polymer and light-absorbing elastomer

Molecular switching and phase transitions have been the primary strategies adopted to impart stiffness tunability to polymers[15]. Owing to the benefits of relatively simple synthetic chemistry, phase transitions, which are classified into glassy-rubbery transition, crystalline-amorphous transition, and anisotropic-isotropic transition, are currently being exploited to develop stiffness-tunable polymers[16,17]. To draw a stiffness map on a thin film structure, glassy-rubbery transition can be a suitable mechanism for the phase transition, as proposed in our design concept, because it allows a rigid polymer to soften under a wide range of elastic moduli in response to a continuous increase in chain mobility with temperature, which is activated by the glass transition temperature. Therefore, herein, we adopted poly(*tert*-butyl acrylate) (P*t*BA), which can alter its stiffness via a glassy-rubbery transition. P*t*BA, which is schematically illustrated in Fig. 2a, was prepared as a thin film via photo-crosslinking after injection casting of a monomer solution of tert-butyl acrylate via capillary action (Supplementary Fig. 1). The resulting P*t*BA film had a rigid-yet-flexible nature (Fig. 2b) and was highly transparent (optical transmittance in the visible-NIR wavelength range (380–1000 nm): >91%; see Fig. 2c).

For photothermal heating, light-absorbing materials such as tosylated PEDOT[18], polydopamine nanoparticles[19], guaiazulene

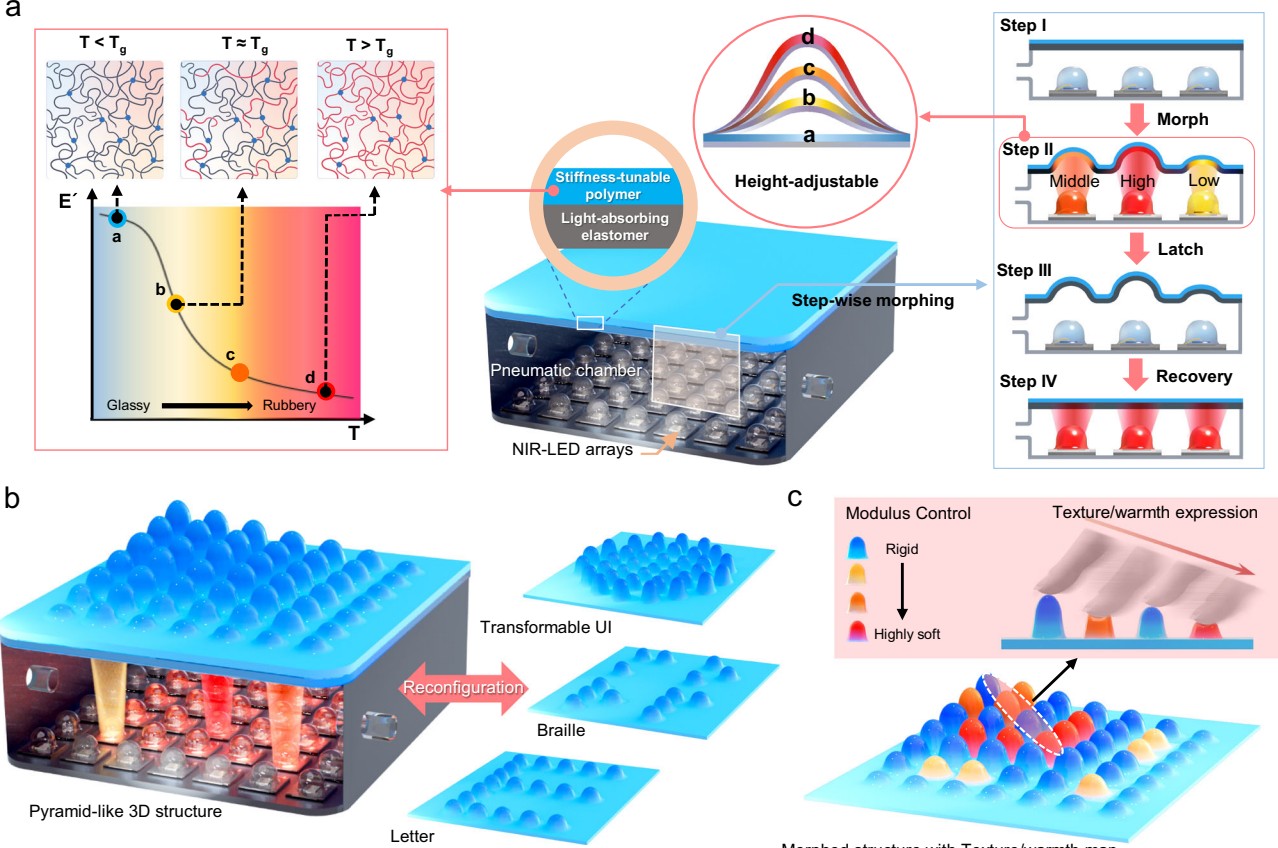

**Fig. 1 | Schematically illustrated concept of the morphable tactile display.**
**a** Structural configuration with an operating mechanism of the morphable tactile display that can produce height-adjustable and latchable deformation via photo-thermally induced stiffness modulation at localized regions of the polymer bilayer under switching of a stationary pneumatic pressure. **b**, **c** Shape-reconfigurable capability of the morphable tactile display, allowing texture with warmth expression on the morphed structure after being latched.

or perylenediimide dye[20], metallic nanowires (e.g., silver nanowires (AgNWs))[21], gold nanorod[22] or gold-porphyrin polymersome[23], liquid metal nanoparticles[24], two-dimensional transition metal carbides/nitrides (MXenes)[25,26], and carbon-based materials[27–30] have been employed. In our preliminary experiments, we confirmed that thin tosylated PEDOT and the AgNWs coating formed on the rigid-yet-flexible polymer possessed an excellent photothermal heating capability, allowing for rapid heating to over 70 °C in response to NIR light irradiation. In particular, tosylated PEDOT enabled localized heating owing to its low thermal conductivity[31]. However, both materials hardly maintained the photothermal heating performance under out-of-plane deformation with a diameter of 4 mm and height of 600 μm, suffering from irreparable cracks during deformation (Supplementary Fig. 2). Therefore, for large, repetitive, and localized photothermally induced deformations, we must consider a material that allows for photothermal and localized heating while simultaneously possessing excellent mechanical robustness with a soft and elastic nature.

As shown in Fig. 2d, we fabricated a photothermal elastomer that was a composite of PDMS with a small amount of GNPs as the nano-filler. A 100 μm thick GNPE membrane was prepared by a sequential fabrication process (Supplementary Fig. 3 and Methods). The Raman spectrum of the GNPs exhibit three characteristic peaks at approximately 1340, 1568, and 2684 cm$^{-1}$, corresponding to D, G, and 2D bands, respectively (Fig. 2e). When compared with the Raman spectra for monolayer graphene and graphite[32,33], the shifts in the G and 2D bands provide clear evidence of GNPs with a few-layer graphene stack. The light absorbance of the GNPEs increased with the GNP content, whereas the light absorbance remained constant irrespective of the wavelength above 300 nm (Fig. 2f). The prepared GNPE membrane with 2.0 wt% GNPs (termed GNPE 2.0) was soft and highly stretchable (Fig. 2g); the elastic nature was maintained even when the GNP content was increased to 5.0 wt% (GNPE 5.0) (Supplementary Fig. 4). Their stress–strain curves revealed that although all GNPEs (GNPE 0.5–5.0) suffered fracture at a lower strain than pristine PDMS, they still retained the intrinsic ductility of the PDMS matrix, allowing elongation ~300%. At the same time, a higher GNP content caused more softening of the GNPEs, accompanied by weakening of their ultimate tensile strength and ductility (Fig. 2h) as well as lowering of the storage modulus to 0.19 MPa under ambient conditions (Supplementary Fig. 5). As shown in Fig. 2i, the GNPEs also exhibited excellent resilience with a small hysteresis that was almost identical regardless of the GNP content. In the case of GNPE 2.0, the hysteresis was 5.18%, comparable with that of pristine PDMS (4.37%), during an elongation–recovery cycle at 100% strain. However, extremely high GNP contents above 10 wt% rarely allowed the GNPE to fully solidify, even after five days of chemical crosslinking (Supplementary Fig. 6). Softening is strongly correlated with the occurrence of carbon-metal interactions between the GNPs and platinum catalysts, resulting from the formation of coordination bonds[34]. We found from our model compound reaction that GNPE 5.0 has 50% lower Si-H consumption than pristine PDMS, indicating a reduction in hydrosilylation reaction for crosslinking of the PDMS prepolymer owing to the interaction of some GNPs in the PDMS matrix with platinum catalysts (Fig. 2j and Supplementary Fig. 7). This indicates that the GNP content available for the mechanically robust GNPE is limited.

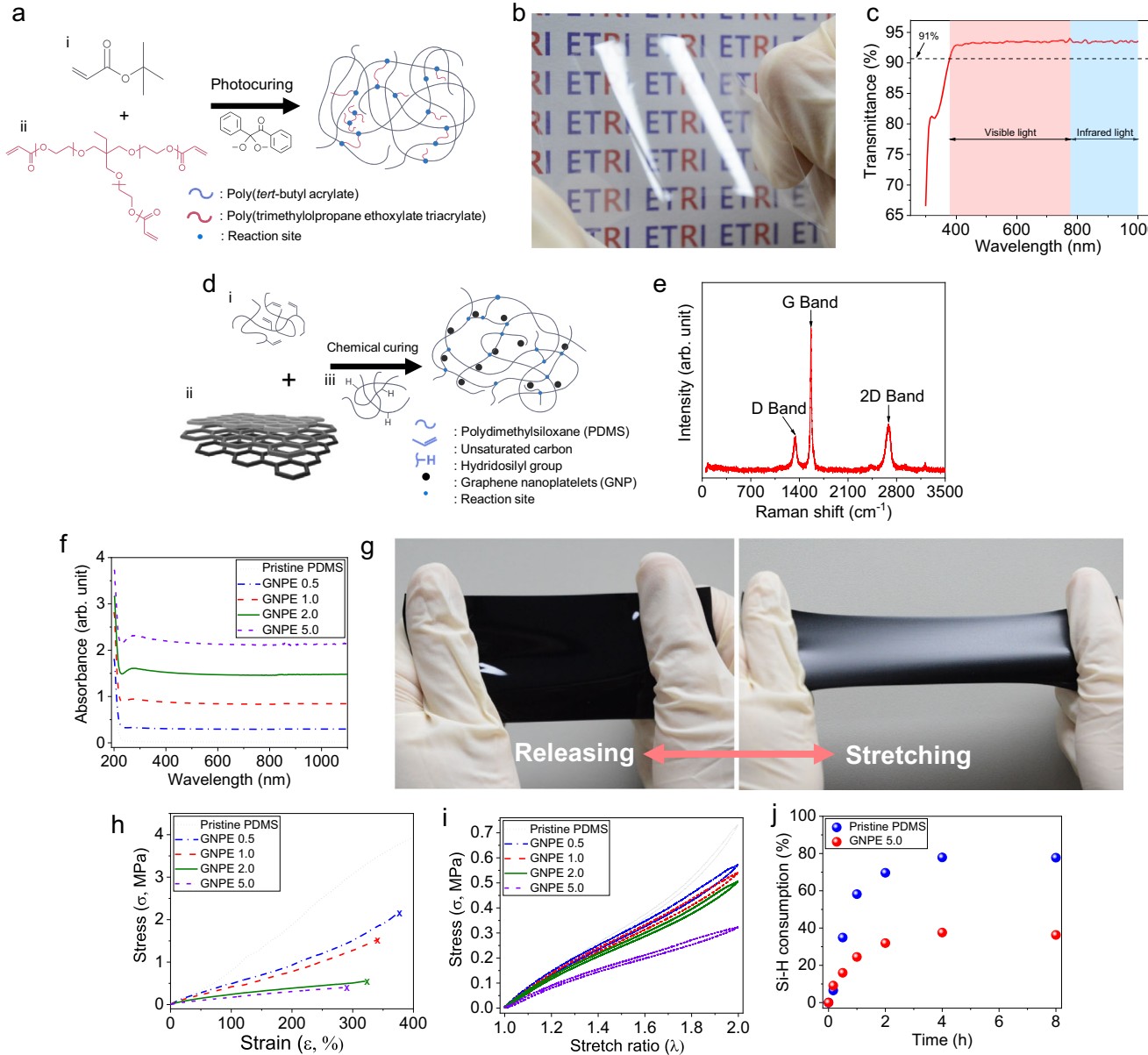

**Fig. 2 | Characteristics of the prepared functional polymers. a** Schematic illustration of chemical structure of P*t*BA. **b**, **c** A photograph and optical transmittance plot of the flexible P*t*BA film. **d** Schematic illustration of chemical structure of GNPE. **e** Raman spectrum of GNPs. **f** Light absorbance spectra of the GNPEs with different GNP wt%. **g** Photographs of the GNPE 2.0 before and after elongation. **h** Strain−stress curves of pristine PDMS and GNPEs at -25 °C. The "x" mark is elongation at break. **i** Elongation−recovery curves of the GNPEs under a stretching-releasing cycle with a stretch ratio of up to 2.0. **j** Comparison of Si-H consumption with reaction time for the pristine PDMS and GNPE 5.0.

## Thermo-mechanics of functional P*t*BA and the GNPE

P*t*BA started to soften at 40 °C, a temperature that promotes glass transition (Fig. 3a). The softening progressed rapidly over a temperature range of 40−70 °C and continued as the temperature increases up to 150 °C. During the rapid softening, the storage modulus of the P*t*BA decreased remarkably from 321.7 to 0.089 MPa at a rate of 10.2 MPa K$^{-1}$. Thereafter, as the temperature increased to 150 °C, the modulus decreased continuously to 2.2 kPa, with a reduction in the change rate to 1.24 kPa K$^{-1}$. In parallel, as shown in Fig. 3b, c, P*t*BA heated to the temperature range that induces rapid softening did not undergo a complete transition to a rubbery state, exhibiting an extremely large hysteresis of over 69% during an elongation−recovery cycle at 100% strain. Although P*t*BA approached a rubbery state as the temperature increases to 150 °C, which caused a reduction in hysteresis to 16.1%, the resulting softened P*t*BA still lacks the resilience required to recover from a deformed state.

Meanwhile, the GNPEs were intrinsically soft, and the storage modulus decreased from 0.61 to 0.19 MPa upon increasing the GNP contents to 5.0 wt% (Fig. 3d and Supplementary Fig. 5). During heating to a temperature of 150 °C, their storage moduli remained stable and exhibited a modest increase corresponding to entropic effect in the rubber elasticity, with thermo-mechanical behavior that higher GNP contents in the PDMS matrix led to a reduction of P*t*BA-GNPE bilayer ΔE′ K$^{-1}$, which is defined as the amount of change in storage modulus per temperature, from 1.76 to 0.64 kPa K$^{-1}$. Notably, the changes in the mechanical properties for the GNPEs were markedly less than those for the P*t*BA between ambient temperature and 150 °C. For the GNPE 2.0, in particular, the ratio corresponding to $E'_{ambient}/E'_{150 °C}$ was only ~0.72, which is ~10$^5$ times smaller than that for P*t*BA (3.06 × 10$^5$), enabling stiffness control depending solely on the mechanical property of the P*t*BA (Inset in Fig. 3d).

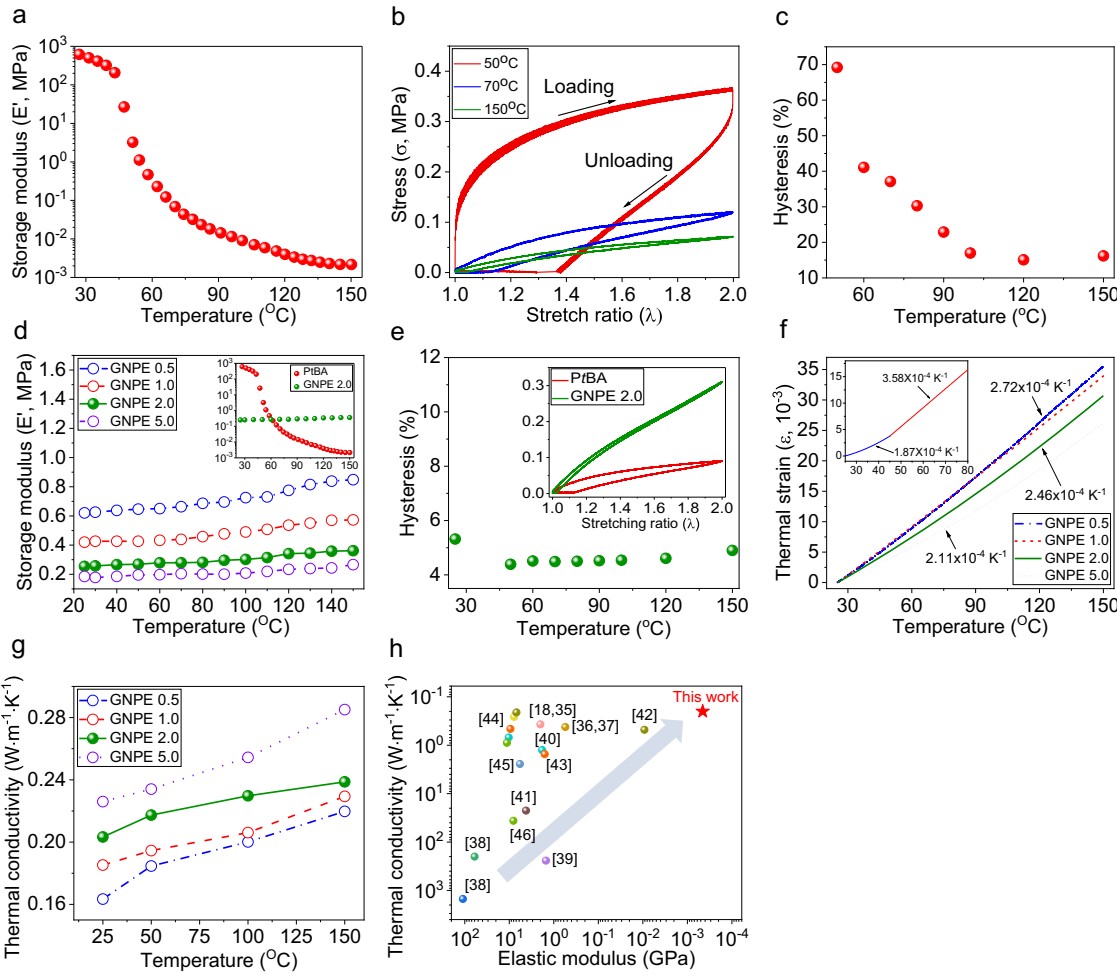

**Fig. 3 | Thermo-mechanical and thermo-electrical characteristics of the prepared functional polymers. a** Temperature-dependent storage modulus of the PtBA. **b**, **c** Elongation–recovery curve and hysteresis of the PtBA achieved at different temperatures under a stretching-releasing cycle with 100% strain, respectively. **d** Comparison of storage modulus profile of the GNPEs (GNPE 0.5–5.0). The inset shows temperature-dependent modulus change in the GNPE 2.0, which is markedly lower than that of the PtBA. **e** Hysteresis of the GNPE 2.0 achieved under the test condition as same as Fig. 3b. The inset is elongation–recovery curve for the PtBA and GNPE 2.0 under heating to 80 °C. **f** Temperature-dependent thermal strain profile of the GNPEs. The linear CTEs of the GNPEs and PtBA are displayed with arrows. For the tests, we prepared thin films of the GNPEs and PtBA with thicknesses of ~200 μm and ~100 μm, respectively. **g** Temperature-dependent thermal conductivity of the GNPEs. **h** Comparison of the thermal conductivity and elastic modulus of the materials enabling photothermal heating.

Furthermore, unlike PtBA, the GNPEs exhibited excellent resilience, enabling recovery from stretching to 100% of their original length with only a small hysteresis (<6%) during the elongation–recovery tests at 27–150 °C (Fig. 3e). Each GNPE possessed a linear coefficient of thermal expansion (CTE) ranging from $2.11 \times 10^{-4}$ to $2.72 \times 10^{-4}\,K^{-1}$ in the temperature range of 25–150 °C. (Supplementary Fig. 8 and Methods); the CTEs were almost constant over the temperature range, exhibiting only a slight decrease with increasing GNP content (Fig. 3f). By contrast, PtBA exhibited two different CTEs when the temperature was increased, increasing significantly from $1.87 \times 10^{-4}$ to $3.58 \times 10^{-4}\,K^{-1}$, due to PtBA becoming rubbery because of glass transition (Inset in Fig. 3f). A comprehensive description of the thermo-mechanical property measurements can be found in the Methods section. We believe that the difference in the CTE (<30%) will hardly cause delamination of the adhesion surface between the PtBA and GNPE layers, as photothermal bimorph actuators have previously been demonstrated by adopting heterogeneous materials with more than a two-fold difference in CTE[18]. Therefore, the GNPE is a versatile material suitable for integration into PtBA thanks to the following features: (i) Qualified light-absorption capability enables photothermal heating of the PtBA layer. (ii) A mild change in the GNPE modulus with temperature allows for control of the amplitude

of deformation, relying exclusively on photothermally induced change in the modulus of PtBA itself. (iii) Its excellent elasticity, independent of temperature condition, complements the weak elasticity of PtBA, which is an obstacle to the complete recovery of the original shape after deformation. (iv) The softness, comparable to that of rubbery PtBA, can help minimize reduction of the achievable deformation while suppressing possible interfacial delamination, which can originate from the mismatch in their stiffnesses during the thermally induced softening of the PtBA.

Moreover, under ambient condition, the thermal conductivity of the GNPEs increased from 0.163 to 0.226 W m$^{-1}$ K$^{-1}$ with increasing GNP contents. Their thermal conductivities increased with rising temperature when heated in the temperature range of 25–150 °C (Fig. 3g). Notably, the GNPE, a polymer composite with carbon fillers, is extremely soft and simultaneously possesses very low thermal conductivity compared to those of other conductive polymers[18,35–37], carbon-based materials[38,39] and carbon-polymer composites[40–46] allowing photothermal heating (Fig. 3h). These unique thermo-electric and mechanical properties are highly suitable for drawing a stiffness map on PtBA through localized photothermal heating without suffering from a significant mechanical mismatch between the photothermal layer and stiffness-tunable PtBA.

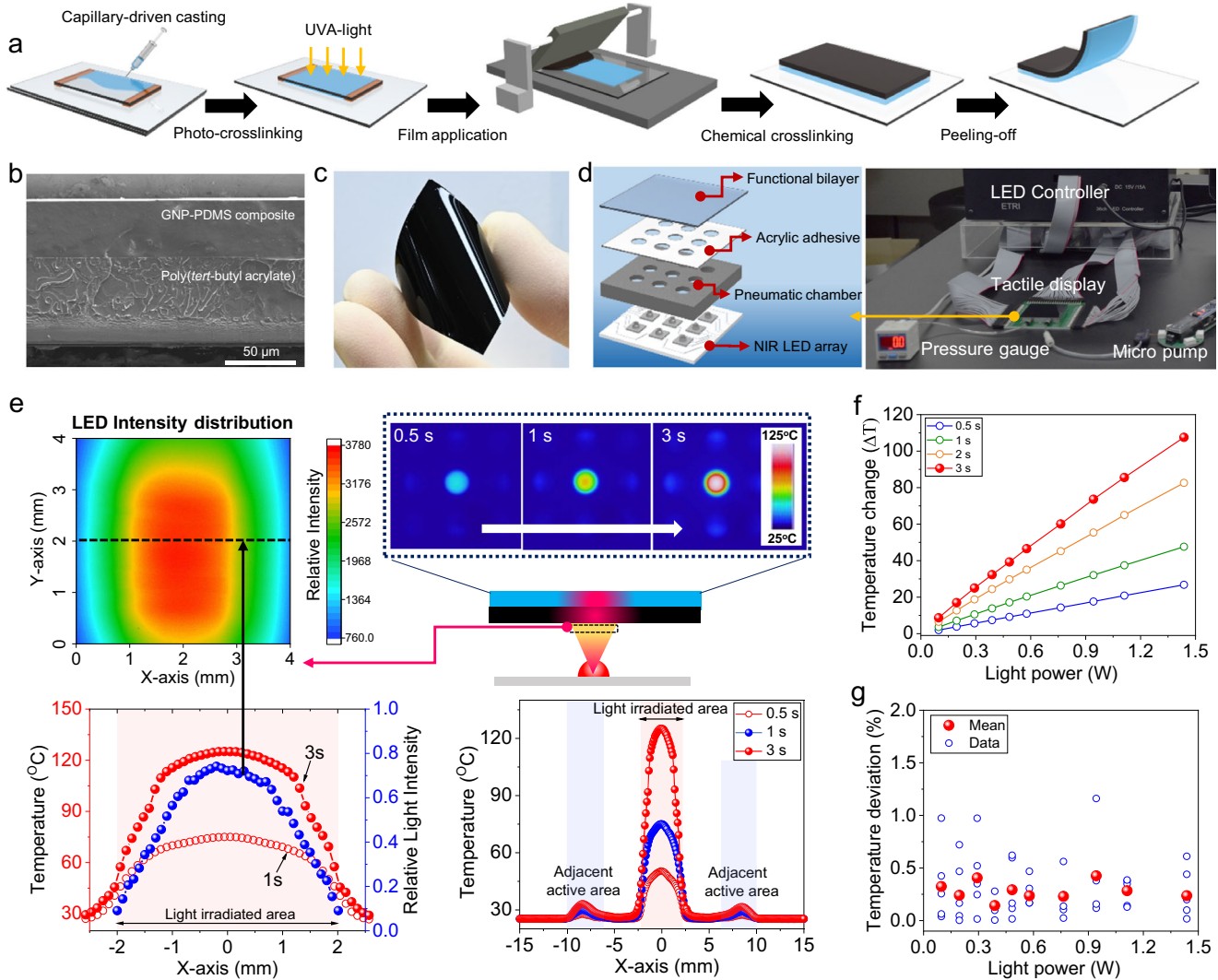

**Fig. 4 | Fabrication and photothermal heating of the PtBA-GNPE bilayer.**
**a** Illustrated fabrication process of the PtBA-GNPE bilayer. **b, c** Cross-sectional SEM image and photograph of the prepared PtBA-GNPE bilayer. **d** Schematic illustration of components to construct a morphable tactile display with a 3 × 3 NIR-LED array. **e** Relative light intensity profile measured at the target area, which is 4 mm distant from an LED operating with a light power of 1.4 W (upper left), comparison of temperature and light intensity distribution at the light-exposed area during photothermal heating (lower left), the thermal image at the light-exposed area with irradiation time (upper right), and position-dependent temperature profile during irradiation of the LED light at the target area (lower right). **f** Change in heating temperature with light power and light exposing time. **g** Deviation of the temperature with light power under light exposure for 3 s (*n* = 6).

## Integration and photothermal heating performance of the PtBA-GNPE bilayer

A bilayer PtBA-GNPE thin film was prepared by casting the liquid GNPE layer onto the bare surface of the photo-crosslinked PtBA (Fig. 4a and Methods). Scanning electron microscope (SEM) revealed that the PtBA-GNPE film formed a stable and smooth interface without structural defects such as cracks and voids, despite being subjected to thermal curing to effectively promote chemical crosslinking in the GNPE at a temperature of 60 °C, which is higher than the $T_g$ of the PtBA (Fig. 4b). The bilayer film was also highly flexible and bendable (Fig. 4c) without occurring delamination at the interface between the PtBA and GNPE 2.0 layers, which exhibited low surface energies of 22.58 and 10.63 mN m$^{-1}$, respectively, as calculated using the Fowkes method[47] (Supplementary Fig. 9 and Note 1); these values are comparable to that of pristine PDMS (10.32 mN m$^{-1}$) due to the inherently hydrophobic nature of PtBA and PDMS as the matrix polymer of the GNPE. The qualified interfacial adhesion between PtBA and the GNPE can be governed by elastic energy rather than potential and surface energy according to the adhesion energy theory proposed by Johnson and Kendall, which indicates that materials with a highly soft nature can

contribute to providing sufficient adhesion energy[48,49]. Therefore, we believe that the chemical crosslinking of the liquid GNPE onto the PtBA layer, which is advantageous for preventing the formation of undesirable air traps at the local interface, also contributed to achieving stable interfacial adhesion.

For the photothermal heating test, the PtBA-GNPE bilayer was integrated into a hardware system designed to supply NIR light via LEDs and pneumatic pressure via a micropump equipped with a pressure gauge (Fig. 4d and Methods). The vertical distance between the LED and bilayer film was set as 4 mm to focus the LED light on the target area with the radiation angle. The spatial design allowed the light emitted from each LED to be focused on each target area (diameter: 4 mm), with the relative light intensity exhibiting a Gaussian distribution (Fig. 4e, upper left). When the LED light was switched on, the light absorbed by the GNPE was converted into thermal energy with a photothermal conversion efficiency of 91.7% (Supplementary Fig. 10, Table 1 and Note 2), resulting in the heating of PtBA only in the light-exposed area. Under a light power of 1.4 W, the achievable temperature for the heated area increases to 125 °C when the light irradiation time is increased from 0.5 to 3.0 s (Fig. 4e, upper right).

Notably, there was no significant thermal diffusion to the outside area, regardless of the temperature, owing to the low thermal conductivity of GNPE compared to other materials, as shown in Fig. 3g. The position-dependent temperature profile at the centerline of the area where the P$t$BA was heated is also similar to the Gaussian-like distribution of the relative light intensity, which is attenuated from the center to the periphery at the exposed area of the LED light (Fig. 4e, lower left), indicating that the GNPE is capable of highly localized heating corresponding to the area-dependent change in the light intensity of the LED even in a small area. The heat conduction to the outside, where the LED light was not directly irradiated, was less than 5 °C, even though the temperature at the light-exposed area reached 125 °C (Fig. 4e, lower right). Additionally, the GNPE possessed the heating capability to modulate the temperature of P$t$BA in proportion to the light power while allowing different settings of the tunable range of the heating temperature according to the light exposure time (Fig. 4f). Furthermore, even under light exposure for 3 s at each light power, the heating performance was still highly reliable, with an extremely low mean deviation (<0.5%) in the temperature (Fig. 4g).

## Photothermal-pneumatic actuation

We prepared a morphable tactile display by integrating the P$t$BA-GNPE bilayer into a pneumatic chamber with a 3 × 3 NIR-LED array. As shown in the conceptual design (Fig. 1), we carried out a cyclic actuation test by sequentially producing out-of-plane deformation at a localized area softened via photothermal heating, latching the deformed state, and recovering to the initial shape through a continuous operating process consisting of four steps: (I) pneumatic pressure application, (II) deformation in response to photothermal heating, (III) cooling, and (IV) removal of pneumatic pressure and second photothermal heating step (Fig. 5a). In Step I, a pneumatic pressure of 6 kPa was applied to the chamber using a micropump system. In Step II, rapid softening of the light-irradiated area of the bilayer by photothermal heating caused out-of-plane deformation. A sharp increase in deformation height was observed when the maximum temperature of the photothermally heated area reached over 55 °C in 0.6 s during the radiation by 1.4-W NIR light. Our tactile display could morph to 90% of the maximum height in 2.4 s and reached 871 μm when irradiated for 3 s. In Step III, the deformed state was latched without any additional energy supply as the softened P$t$BA layer regained its intrinsic rigidity upon cooling to below 35 °C. After reducing the pneumatic pressure to atmospheric pressure, the second photothermal heating step in Step IV enabled the deformed state to be restored to a planar shape. In terms of the actuation time, this performance can be improved using several strategies. First, a stronger and shorter light pulse can be applied, which causes a steeper increase and decrease in the surface temperature and modulus, respectively, while consuming the same amount of energy. It is possible to optimize the storage modulus of the functional material depicted in Fig. 3a, aiming for a steeper decreasing slope at lower temperature. A softer surface will result in quicker inflation of height deformation. For a faster-rising response, Step I and II can be combined by applying air pressure and NIR light simultaneously; however, this is accompanied by a slight reduction in maximum height. The height recovery in Step IV predominantly relies on the spring force of the bilayer film, resulting in a slower response than the rising response. This recovery process can be expedited by employing negative air pressure.

The latched shape achieved through Step III could withstand a load corresponding to the reaction force of a typical keyboard (0.5 N) without any significant structural distortion (height loss: ~46 μm) (Fig. 5b, Supplementary Fig. 11, and Note 3). A holding force of 2.4 N was required for pressing down on the latched shape by 400 μm, while a load of 6.2 N, which is significantly higher than the 1–1.5 N reported in previous studies[7,8], was required for the shape to collapse completely (Supplementary Fig. 11). The holding force, which is ~80-fold higher than the detection threshold of a human fingertip (~30 mN[50]), is comparable to the force of pin-type tactile display that was determined for haptic exploration of visually impaired people[51]. Taking advantage of the excellent stiffness tunability on the P$t$BA, the deformation height of the latched shape was photothermally tunable to unprecedented multi-levels, ranging from 10.3 to 870 μm with only on/off control of constant and low pneumatic pressure (Fig. 5c, d). The deformation heights achievable via the photothermal-pneumatic stimulus were highly reproducible with a small mean deviation of <1% when the light power exceeded 0.5 W (Fig. 5e). Below 0.5 W, a relatively high deviation, though still small at only a few percent, occurred, which correlated with unstable softening behavior arose from a vigorous increase in chain mobility of P$t$BA in the temperature range of 40–60 °C.

Figure 5f shows the deformation height of the P$t$BA-GNPE bilayer in response to the temperature increase induced by light irradiation. To estimate the deformation behavior of the P$t$BA-GNPE bilayer, we performed thermo-mechanical simulations and included the results in Fig. 5f (see Supplementary Note 4). The experimentally measured deformation height showed a steep increase at temperatures above 40 °C; the deformation profile was in good agreement with the simulation results. The slight discrepancies observed in the displacement profiles are attributed to heat conduction losses to the contacting substrate along the circumference of the P$t$BA-GNPE bilayer. This temperature-dependent deformation behavior is attributed to the differences in material properties between the GNPE and P$t$BA layers. As shown in Fig. 3, properties such as the elastic modulus, CTE, and thermal conductivity of the GNPE and P$t$BA layers were highly temperature-dependent. In addition, the GNP content significantly affected the modulus of the GNPE layer. Therefore, we performed further simulations (see Supplementary Table 2 and Fig. 12) to investigate the deformation behavior of the proposed GNPE-P$t$BA bilayer under different material properties. These simulations provided valuable insights into the photothermal deformation behavior of the P$t$BA-GNPE bilayer and a background for optimizing its actuation performance.

Meanwhile, we performed a durability test of the photothermal-pneumatic actuation under stepwise deformation-recovery cycles. During the first 10 continuative actuation cycles, the activated area exhibited reversible deformation-recovery behavior, with a consistent deformation rate (700 μm s⁻¹) and recovery rate (380 μm s⁻¹), which were calculated by differentiating the deformation-recovery plot with time (Fig. 5g). It was also observed that the photothermal-pneumatic actuation was durable and stable with a small deviation (<5%) even during 5000 actuation cycles (Fig. 5h). Reversible actuation, with the assistance of the elastic nature of the GNPE layer, was successfully enabled because change in deformation height of up to 1.42 mm rarely caused degradation of the reachable temperature via photothermal heating even under the same light power despite increasing distance between the light source and P$t$BA-GNPE bilayer in response to increase in the deformation height (Supplementary Fig. 13). Overall, our morphable actuator could produce a deformation height that could be controllable at multiple levels with the light irradiation power and was both reproducible and durable, surpassing the performances of previously reported morphable actuators.

## Multi-functional morphable tactile display

Finally, we prepared a morphable tactile display device by integrating a large-area P$t$BA-GNPE bilayer film in a pneumatic chamber with a 6 × 6 NIR-LED array (Fig. 6a). We conducted tests to demonstrate that our photothermal-morphable tactile display could generate physical shapes and reconfigure them into other shapes by modulating the deformation heights of 36 areas in response to light signals from individual LEDs in a 6 × 6 array under constant pneumatic pressure. Thanks to the excellent light-absorbing properties of the GNPE with

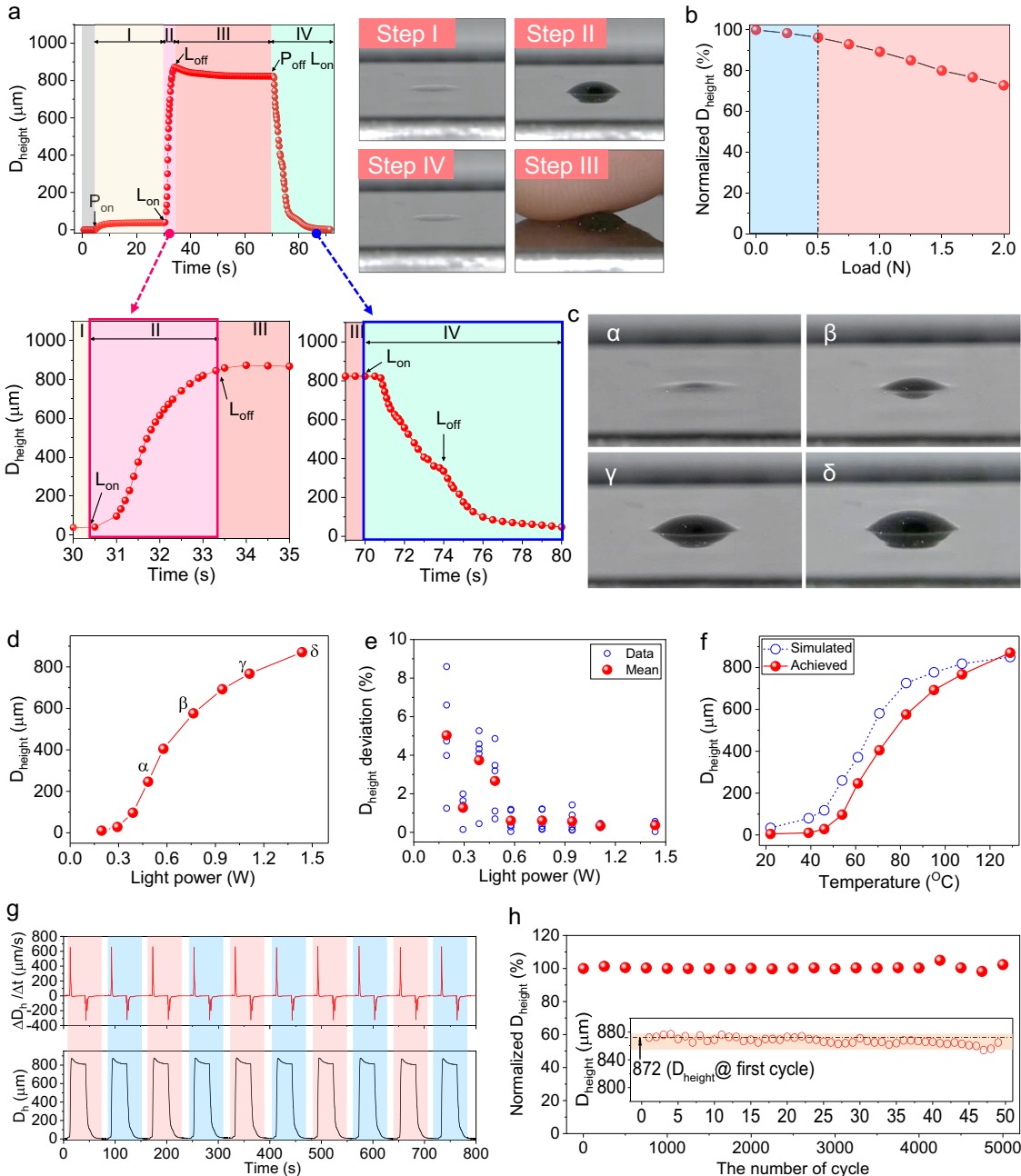

**Fig. 5 | Photothermal-pneumatic actuation performance. a** Change in deformation height and photographs of a morphed area at each step during a stepwise deformation-recovery cycle. **b** Load-dependent change in deformation height after being latched in step III. **c** Photographs of a target area morphed to have different deformation heights by controlling the light power. α, β, γ, and δ correspond to 0.4, 0.8, 1.1, and 1.4 W, respectively. **d**, **e** Change in deformation height on the morphed area with light power and its deviation that was achieved through five repeated tests at each light power condition. **f** Comparison of experimentally measured deformation height and simulation result at each temperature condition. **g** Deformation-recovery plot with deformation and recovery rate during 10 continuative actuation cycles. **h** Durability test of the photothermal-pneumatic actuation under 5000 stepwise deformation-recovery cycles. The inset shows the maximum deformation height achieved during 50 continuous actuation cycles.

extremely low thermal conductivity, 36 areas on the P$t$BA-GNPE bilayer were locally heated, even during the simultaneous operation of all NIR-LEDs (Fig. 6b). Each area was independently morphed into a three-dimensional curved shape with different deformation heights. By applying higher pneumatic pressure (18 kPa instead of 6 kPa), the maximum expressible deformation height could be amplified to 1420 μm while maintaining reversible actuation without increasing the light power (Fig. 6c); simultaneously, precise fourteen-step control of the deformation height could be realized (Fig. 6d). In particular, as shown in Fig. 6e, f, which summarize its tunable steps with the

maximum deformation height and the holding force of previously reported tactile displays[7–10,52–58], respectively, the tactile display developed in this work (red star) allowed for a wide range of controllable deformations with a number of tunable steps comparable to electrical tactile displays, and possessed the highest load-bearing capability (holding force: 2.4 N).

Moreover, the surface morphing capability enabled not only the physical expression, erasure, and rewriting of letters, such as Braille alphabets and English alphabets (Fig. 6g, h and supplementary Movie 1) but also the generation of the topology of a three-dimensional

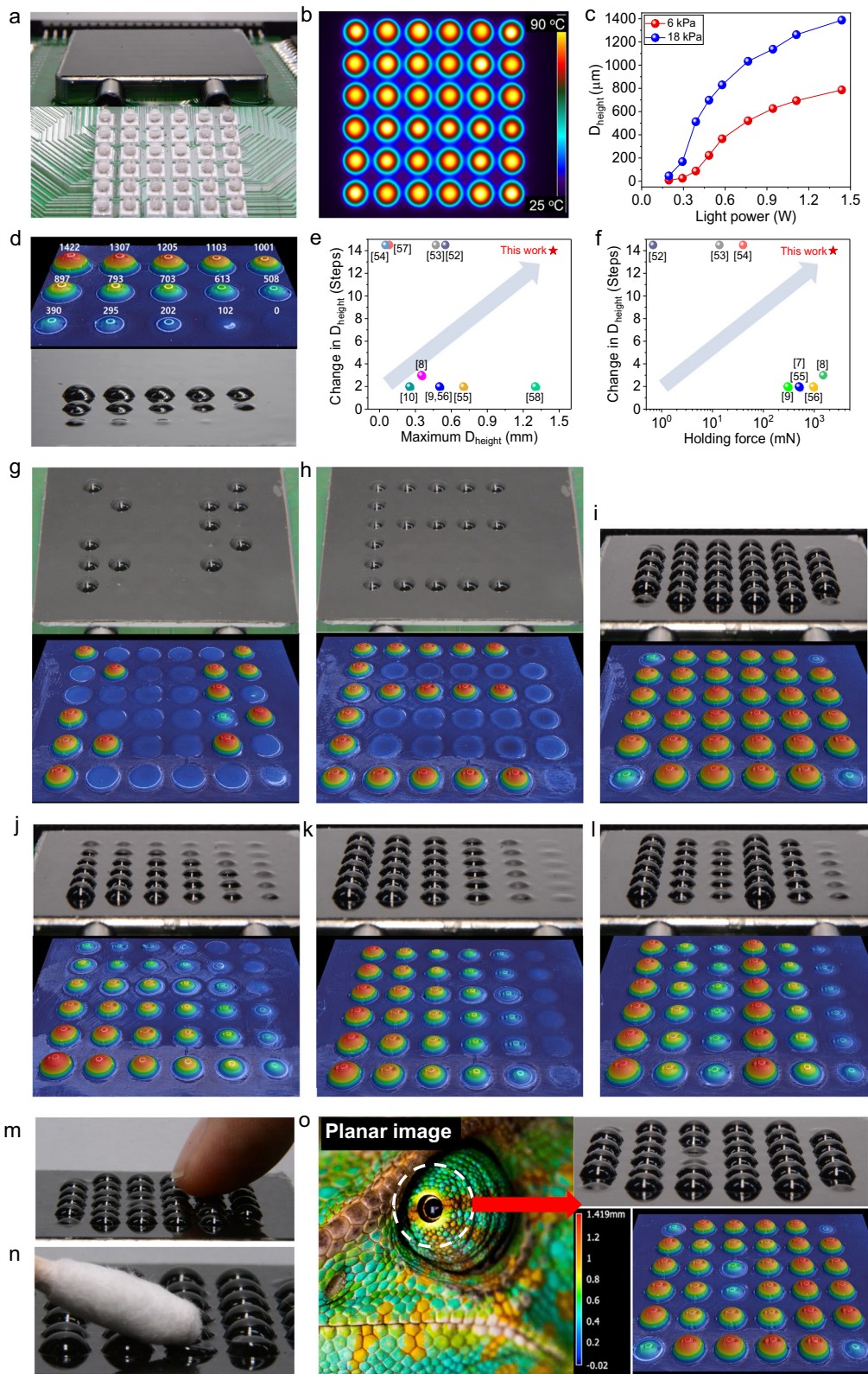

**Fig. 6 | Reconfigurable topological expression capability of the morphable tactile display. a** Photograph of the morphable tactile display device integrated with a 6 × 6 NIR-LED array. **b** Thermal image of 36 areas photothermally heated during operation of all NIR-LEDs. **c** Change in the maximum expressible deformation height of the morphed areas with light power under two different pneumatic pressure conditions. **d** 3D profiler image of the areas morphed into curved shapes with different deformation heights. **e, f** Comparison of tunable steps with the maximum deformation height and holding force. **g–l** Photographs with 3D profiler images of our tactile display that physically expressed not only Braille alphabets for "ETRI" and alphabet "E" but also 3D topologies of a control knob, a pyramid, and different contours, respectively. **m, n** Photographs showing pressurized finger touch on the morphed surface after being latched and capability locally adjusting softness with warmness on the morphed surface, respectively. **o**, Photograph with 3D profiler image of textured 3D topology, mimicking the skin of a chameleon eye.

structure and its reconfiguration into other structures, such as a control knob, pyramid, and different contours (Fig. 6i–l and supplementary Movie 2), withstanding the pressurized rubbing force with the finger (Fig. 6m). Furthermore, with this latchable topological expression, our tactile display could locally adjust softness with warmth applied on the 3D topologies, where the morphed areas would have cooled and recovered their original stiffness in response to the secondary light-triggered stiffness change in specific areas, without undergoing pressure reduction (Fig. 6n). The morphed areas maintained their shape regardless of light exposure, and the light-exposed areas, unlike the others, could only be pressed down when touched or rubbed with fingers because of the resulting softening, allowing the areas to be more pressurized and warmable as the applied light power increased (Supplementary Fig. 14). Because of this functionality, our tactile display could generate a textured 3D topology, such as mimicking the skin of the chameleon eye (Fig. 6o and supplementary Movie 3).

## Discussion

In this study, we developed a functional polymer bilayer integrating the light-absorbing GNPE into the stiffness-tunable P*t*BA that could photothermally construct a rewritable stiffness map. Complementing the insufficient elasticity of P*t*BA, particularly in the temperature range for a sharp rigid-to-rubbery transition, the GNPE with extremely low thermal conductivity allowed P*t*BA to be heated in a confined area irradiated with light. Highly localized heating through temperature control in proportion to light power allowed height-adjustable deformation (tunable steps: 14) at each light-exposed area under constant pneumatic pressure. The deformation was reversibly latched-to-release in response to the stepwise control of the photothermal-pneumatic stimulus, accompanied by a consistent deformation-recovery rate.

Exploiting this programmable deformation in confined areas, our morphable tactile display could generate a shape and change into another shape, resulting in the expression of not only 3D physical information such as Braille or the English alphabet but also geometrical topology with height variation. Moreover, by implementing a secondary light-triggered stiffness change in partially or fully morphed areas where the deformed shape was latched, it could also generate a texture with warmth on the 3D topology when touched or rubbed with fingers.

We believe that our morphable tactile display, which adopts an optical system with high-integration and large-area LED arrays, can open a promising route toward revolutionary electronic devices, such as, rewritable informative devices providing Braille and unsymbolized information for visually impaired people, transformable UIs for automobiles, physical communion in metaverse space, 3D identification, and interactive educational apparatus providing high-dimensional information of objects.

## Methods

### Preparation of the P*t*BA, GNPE, and P*t*BA-GNPE bilayer

*Tert*-butyl acrylate (*t*BA) and 2,2-dimethoxy-2-phenylacetophenone (DMPA) were purchased from Sigma-Aldrich, and Trimethylol-propane ethoxylate triacrylate was obtained from Sartomer Company (SR9035). For P*t*BA, a liquid monomer, which consists of 100 parts of *t*BA, 0.5 parts of DMPA as photo-initiator, and 0.5 parts of SR9035 as crosslinker, was prepared by mixing them under magnetic stirring at ambient condition. The monomer solution was injected into a 100 μm gap between two glass plates, which was formed using a couple of polyimide adhesive spacers, and photo-crosslinked by irradiation of UVA-light (wavelength: 365 nm, power: 1 kW) for 10 min under a continuous flow of high-purity nitrogen. After removing the cover glass plate, a P*t*BA layer on the bottom glass plate was dried in a vacuum oven at 60 °C for 6 h to eliminate unreacted homopolymer, soaked into deionized water, peeled off from the glass plate, and dried by blowing with nitrogen gas.

Graphene nanoplatelets (GNPs) with a <2 μm size and a few nm thickness and hexane were purchased from Sigma-Aldrich. A base compound and curing agent of PDMS were purchased from Dow corning (Sylgard 184). A liquid GNPE was prepared via a stepwise process: diluting PDMS prepolymer with hexane (prepolymer/hexane: 0.033 g mL$^{-1}$) in a glass beaker, mixing with GNPs under mechanical stirring, dispersing GNPs in PDMS matrix under sonication (power: 150 W) at ~70 °C as evaporating the solvent molecules for 3 h, and adding curing agent to the emulsion composed of PDMS prepolymer and GNPs with a 15:1 weight ratio of prepolymer to curing agent. After degassing in a vacuum oven, the liquid GNPE was cast on a glass plate using a Zehntner 2300 film applicator and solidified as a thin film via a chemical crosslinking process in a heating oven at 70 °C for 12 h.

A P*t*BA-GNPE bilayer was prepared via a stepwise process: degassing from liquid GNPE for 20 min, casting it onto a naked surface of P*t*BA layer on a glass substrate, crosslinking at 60 °C for 24 h, soaking into a deionized water to facilitate delamination of the P*t*BA-GNPE bilayer from glass substrate, and drying by blowing with nitrogen gas after peeling off from glass substrate.

### Construction of photothermal-pneumatic morphable display device

A morphable tactile display device was fabricated by integrating a P*t*BA-GNPE bilayer film, which has a size of ~0.2 × 40 × 40 mm, into a metal pneumatic chamber with a 6 × 6 NIR-LED array. The NIR-LEDs (Luminus SST-10-IRD-B50-U940, peak wavelength: 940 nm) were mounted on a metal PCB board with a 5-mm pitch. To airtightly seal an interface between the pneumatic chamber and metal PCB board, liquid silicone elastomer (Sylgard 184, Dow Chemical) was coated on the interface, and cured in a drying oven at 60 °C for 2 h. Finally, A prepared P*t*BA-GNPE bilayer, placing the GNPE side down toward the LEDs, was attached to the surface of the pneumatic chamber using a 130 μm thick adhesive layer (3 M VHB F9469PC), which was perforated via laser cutting system (Universal Laser system). The perforation could help avoid integration of the adhesive layer into active areas that morph in response to the photothermal-pneumatic stimulus and contributed to preventing a reduction in the achievable deformation height under the same pneumatic pressure. A micropump equipped with a pressure gauge (SMC ZSE30AF), which was connected to the pneumatic chamber through small nozzles on either side, was used for on/off control of pneumatic pressure.

### Thermo-mechanical measurement

Thin rectangular specimens of the P*t*BA and GNPEs (GNPE 0.5–5.0), which have a thickness of ~100 μm, were prepared by following ASTM standard D882. Their mechanical and thermo-mechanical properties were measured by a TA instruments RSA-G2 solids analyzer. Stress–strain curves were obtained by stretching the specimens to their breaking points at a temperature of ~25 °C under a stretching rate of 0.005 mm s$^{-1}$ (Fig. 2h). Elongation–recovery curves were obtained in the temperature range of 25–150 °C by stretching and releasing the specimens under the stretching rate of 0.005 mm s$^{-1}$ after stabilization at each target temperature, which was reached under a heating rate of 5 K min$^{-1}$, for 10 min (Figs. 2i and 3b). From each curve, hysteresis, which is defined as a percent ratio of the area between loading and unloading curves to the area of the loading curve, was calculated by $(A_{\text{loading}} - A_{\text{unloading}}) A_{\text{loading}}^{-1}$. Storage moduli with temperature were measured by applying a dynamic strain of 5% with a frequency of 1 Hz during continuous heating to a temperature of 150 °C under a heating rate of 5 K min$^{-1}$ (Fig. 3a, d). The reported storage moduli were the average of 10 values, which were measured repeatedly at each temperature condition. Temperature-dependent thermal strains of the P*t*BA and GNPEs (8 mm long, 3 mm wide) were measured using TA instruments Discovery TMA 450 by applying a load of 1 mN. Linear CTE of the polymers was calculated from slope of

their thermal strain profiles, which were achieved during second heating continued after first cyclic heating-cooling with a constant rate of 2 K min$^{-1}$ to remove thermal history in the polymers (Fig. 3f). Thermal conductivity of the GNPEs (GNPE 0.5–5.0) was calculated from their thermal resistance at four different temperature conditions (ambient temperature–150 °C), which was measured by a TA Instruments DTC-300, as simultaneously providing a constant pressure of 30 psi by following ASTM E 1530 (Fig. 3g). Three disc-shaped specimens for each GNPE (diameter: 25 mm, thickness: 500 μm) were used for the measurements.

## Optical measurement

Optical transmittance of the P*t*BA film (thickness: ~100 μm) and light-absorption spectra of the GNPEs (thickness: ~100 μm) were measured by a Shimadzu UV-2600 spectrometer (Fig. 2c, f). Raman spectrum of GNPs was obtained from a HORIBA LabRAM HR Evolution Raman spectroscope using a Laser (wavelength: 532 nm) (Fig. 2e). Cross-sectional image of the P*t*BA-GNPE bilayer was taken by a Sirion 600 field emission scanning electron microscope (Fig. 4b). Wettability of the P*t*BA and GNPEs was evaluated by measuring contact angle via a KRÜSS DSA 25 S drop shape analyzer. Deionized water and diiodomethane were used as test liquids. A customized multi-channel current controller, which is connected to a metal PCB where NIR-LEDs were mounted, was used to adjust the operating current and duration of each LED. Temperature distribution on the surface of the bilayer during photothermal heating in response to light irradiation using NIR-LEDs was measured by a FLIR A6781 MWIR thermal camera (50 frames per second) (Figs. 4e–g and 6b). Relative light intensity of the NIR-LED light was measured at the exposed area (diameter: 4 mm), which is 4 mm distant from the NIR-LED, using a Spiricon SP620U beam profiling camera (Fig. 4e).

## Computational simulation

A numerical analysis was conducted using the finite element method with ANSYS Workbench 2021 R1 (Fig. 5f). The Large Deflection option was enabled in the static structural analysis. To improve the convergence of the simulation, sub-stepping techniques were used by setting 20 initial sub-steps; a minimum of 10 and a maximum of 100 sub-steps. In addition, a multi-zone meshing technique with 363,432 nodes and 65,618 hexahedral elements was used to optimize the representation of the physical system and accurately capture complex deformations.

## Morphing performance test

Three-dimensional shapes and deformation heights of the morphed areas were measured by a KEYENCE VR-3000 3D profiler (Fig. 6). Change in the deformation height at an active area (diameter: 4 mm) during cyclic photothermal-pneumatic actuation was measured by a Polytec PSV-500 laser scanning vibrometer (Fig. 5d–h). Holding force was measured by placing different weights on the morphed structure and then pressing down on the morphed shape using an Optic Focus MOXYZ-02 motorized vertical stage with a DACELL UMI-G500 load cell and a cylindrical contact tip made of acrylic resin (diameter 6 mm). Force data was collected using a control PC with an NI USB-6351 DAQ (Fig. 5b).

## Data availability

Data supporting the findings of this manuscript can be found in the Supplementary Information and available from the corresponding author upon request. Source data are provided with this paper.

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

## Acknowledgements

This work was supported by the internal grant of the Electronics and Telecommunications Research Institute (ETRI) (23YB1700, Development of light-driven three-dimensional morphing technology for tangible visuo-haptic interaction).

## Author contributions

S.Y. and I.H. contributed to the experimental design and data analysis. I.H., S.M., J.-H.Y., S.P., and M.C. carried out the experiments. H.J.K. and T.K. performed the numerical simulations. S.Y. and I.H. wrote the manuscript. Q.P. contributed to the analysis of materials with suggestions. All authors contributed to revising the manuscript. S.Y. supervised the study.

## Competing interests

The authors declare no competing interests.
