## [Peer Review File · Nature Communications]

REVIEWER COMMENTS

Reviewer #1 (Remarks to the Author):

In the manuscript titled “A height-renderable morphable tactile display enabled by programmable modulation of local stiffness in photothermally-active polymer”, the authors present their work on the development of a photothermal tactile/haptic display array. This achievement is realized through a systematic investigation of two material systems. The work demonstrates excellent research rigor and is of interest to researchers in tactile sensing, haptic feedback, multi-modal human-machine interfaces, and extended reality (XR) devices/systems. The authors should address the following points to make it suitable for publication:

1) Please use SI (Système International) standards with units. Please use periods between derived units expressed as products of units.

Eg. mm s-1 should be mm.s-1 and so on for other units.

2) The authors have tested it for a relatively low number of cycles. Claims such as “is highly durable and stable even after ten cyclic actuation”, “deformation height profile and the maximum deformation height achieved at each cycle during around fifty cyclic actuation”, “hardly causes degradation” is a bold claim. For multimodal interaction such a device should endure tens of thousands of cycles.

3) “the highest load bearing capability (holding force 2.4 N)”: The authors should comment on how this compares to the range/lime of human tactile sensing.

4) “o, A textured 3D topology, mimicking the skin of a chameleon eye [For the photograph of the chameleon , which was obtained from GettyImages Bank, you may not distribute or resell the content without permission].”: I recommend the authors consider using another open source or creative commons image instead of this particular chameleon image, as this work will likely be cited and used in several review articles/books, and using such an image may impede further sharing of this work. This could be moved to supplementary material.

5) Page 22: “which was perforated via laser cutting system”. What is the reason for the perforation?

6) I recommend the authors to add a fabrication schematic diagram for further repeatability of this work by other researchers.

7) The authors should aim to get some support to improve the quality of English-language writing of the manuscript. The below are just representative. The entire article requires careful reading and revision.

Some of the representative discrepancies include:

a) The first sentence in the abstract needs to be revised.

b) “The morphable tactile display is composed of “the” heterogeneous polymer structure integrating stiffness. Use of “the” in the first use.

c) This must be revised. “Tactile displays have been developed to convey textual information for the visually impaired and to enhance the user experience via multimodal interaction.”

d) This should be revised: “Recently, height-adjustable tactile displays have introduced in the form of tactile pin arrays”

e) “the pins to wobble. In addition, the rigid mechanical structure is difficult to make flexible enough to”

f) “et al.” must be italicized.

g) Page 2: “expression is required for both for the visually impaired and multisensory tangible interaction.”

h) Page 3: “on the morphed structure by photothermally induced in stiffness modulation at localized regions of the polymer bilayer.”

i) Page 5: “elongation exceeding 300%. At the same time, higher GNP content causes more softening of the GNPEs, accompanied by a weakening their ultimate tensile strength and ductility”

j) Page 7: “higher GNP contents in the PDMS matrix reduces of”

k) Page 22: “were established on a metal PCB with”, “For morphing actuation tests at thirty-six areas of the PtBA-GNPE bilayer, we prepared”

Reviewer #2 (Remarks to the Author):

This manuscript reports a photothermally-modulated height-renderable morphable tactile display which can tune both stiffness and height of the 3D Gaussian protrusion by introducing graphene nanoplatelet (GNP)-polydimethylsiloxane (PDMS) composite elastomer (GNPE)/photo-crosslinked poly(tert-butyl acrylate) (PtBA) bilayer. The two major properties were controlled by photothermal heat and pneumatic pressure, respectively. Although the materials are well known, there is still a lot of research being proceed and good application demonstration have been shown. Therefore, this manuscript has sufficient originality and novelty. Detailed analysis and experiment methods are easy to understand. I suggest some additional comments and questions below.

1. I think the authors could add more latest references to photothermal related papers in the literature for readers.
2. The protruding deformation in the figures and videos seems large that it is difficult to touch the multiple protrusions with the index finger. Optionally, is there an easy way to reduce the size of the protruding deformation to similar to the size of Braille (1-2 mm)?
3. In figure 5a (step II), it seems that the surface temperature of the bilayer can rise upto 125 oC. How long does it take to cool down to room temperature (can touch fingers)?
4. It would be better to provide a height vs. temperature profile in the blank space in figure 3.
5. The authors compared the thermal conductivity and elastic modulus of the materials enabling photothermal heating. How about comparing the response time and protrusion height (or size) with other displays? And it would be better to summarize these comparison values in a table in supplementary information.
6. Would you provide a photothermal conversion efficiency of graphene nanoplatelet (GNP)-polydimethylsiloxane (PDMS) composite elastomer?

REVIEWER COMMENTS

Reviewer #1 (Remarks to the Author):

In the manuscript titled “A height-renderable morphable tactile display enabled by programmable modulation of local stiffness in photothermally-active polymer”, the authors present their work on the development of a photothermal tactile/haptic display array. This achievement is realized through a systematic investigation of two material systems. The work demonstrates excellent research rigor and is of interest to researchers in tactile sensing, haptic feedback, multi-modal human-machine interfaces, and extended reality (XR) devices/systems. The authors should address the following points to make it suitable for publication:

1) Please use SI (Système International) standards with units. Please use periods between derived units expressed as products of units. Eg. mm s⁻¹ should be mm.s⁻¹ and so on for other units.

[Authors]: Thank you for kind comment. We carefully reviewed our manuscript including figures (units for x-axis and y-axis) and corrected the derived units, which were not in accordance with SI standards. In our revised manuscript, the modified units were highlighted by using yellow color.

2) The authors have tested it for a relatively low number of cycles. Claims such as “is highly durable and stable even after ten cyclic actuation”, “deformation height profile and the maximum deformation height achieved at each cycle during around fifty cyclic actuation”, “hardly causes degradation” is a bold claim. For multimodal interaction such a device should endure tens of thousands of cycles.

[Authors]: Thank you for kind comment. We would agree with reviewer’s opinion that around 50 cyclic actuation tests are not enough to claim highly durable performance of our morphable actuator. As carefully considering reviewer’s opinion, we markedly increased the number of cyclic actuation tests from 50 to 5,000. We could observe that the photothermal-pneumatic actuation was durable and stable with a small deviation (<5%) during the 5,000 cycles. In our revised manuscript, we modified description related to the durability test and Fig. 5h including an inset, which was Fig. 5h (upper) in our original manuscript, showing the maximum deformation height during the first 50 actuation cycles. The modified parts were highlighted by using yellow color.

3) “the highest load bearing capability (holding force 2.4 N)”: The authors should comment on how this compares to the range/lime of human tactile sensing.

*[Authors]: Thank you for kind comment. Force detection threshold of fingertip has been reported to be ~30 mN [ref. *]. A study on flexible haptic display reported that a holding force of over 100 mN was required to be easily perceivable by end-user [ref. **]. In particular, to examine psychophysical factor that is required to develop a tactile device for visually-impaired people, an experiment has determined the force to be 2.35 N for normal haptic exploration [ref. ***]. These previous studies support that our morphable actuator, with a holding force of 2.4 N that is ~80-fold higher than the force detection*

threshold, can allow the normal haptic exploration for visually-impaired people.

[] King, H. H., Donlin, R. & Hannaford, B. Perceptual thresholds for single vs. Multi-Finger Haptic interaction. IEEE Haptics Symposium, Waltham, MA, USA, pp. 95-99 (2010).*

*[**] Besse, N., Rosset, S., Zarate, J. J. & Shea, H. Flexible Active Skin: Large Reconfigurable Arrays of Individually Addressed Shape Memory Polymer Actuators. Adv. Mater. Technol. 2, 1700102 (2017).*

*[***] Vidal-Verdú, F. & Hafez, M. Graphical tactile displays for visually-impaired people. IEEE. Trans. Neural Syst. Rehabil. Eng. 15, 119-130 (2007).*

4) “o, A textured 3D topology, mimicking the skin of a chameleon eye [For the photograph of the chameleon, which was obtained from GettyImages Bank, you may not distribute or resell the content without permission].”: I recommend the authors consider using another open source or creative commons image instead of this particular chameleon image, as this work will likely be cited and used in several review articles/books, and using such an image may impede further sharing of this work. This could be moved to supplementary material.

[Authors]: Thank you for kind suggestion. As considering further sharing of this work, we replaced the copyrighted chameleon image (Fig. 6o) with an analogous image using for free.

5) Page 22: “which was perforated via laser cutting system”. What is the reason for the perforation?

[Authors]: Thank you for question. As illustrated in Fig. 4d, the metal pneumatic chamber was perforated to provide local areas of the PtBA-GNPE bilayer with a pneumatic pressure. The perforation could help avoid integration of the adhesive layer into active areas that morph in response to the photothermal-pneumatic stimulus. Therefore, it contributed to preventing reduction in the achievable deformation height under the same pneumatic pressure. As carefully considering the question from the reviewer, we additionally described the aim of the perforation in Methods to assist readers in understanding. The added sentence was highlighted by using yellow color.

6) I recommend the authors to add a fabrication schematic diagram for further repeatability of this work by other researchers.

[Authors]: Thank you kind comment. In supporting information (Please see Supplementary Figs. 1 and 2), we presented the schematic diagrams, showing stepwise fabrication process of the PtBA and GNPE layers, respectively.

7) The authors should aim to get some support to improve the quality of English-language writing of the manuscript. The below are just representative. The entire article requires careful reading and revision. Some of the representative discrepancies include:

a) The first sentence in the abstract needs to be revised.

b) “The morphable tactile display is composed of “the” heterogeneous polymer structure integrating

stiffness. Use of “the” in the first use.

c) This must be revised. “Tactile displays have been developed to convey textual information for the visually impaired and to enhance the user experience via multimodal interaction.”

d) This should be revised: “Recently, height-adjustable tactile displays have introduced in the form of tactile pin arrays”

e) “the pins to wobble. In addition, the rigid mechanical structure is difficult to make flexible enough to”

f) “et al.” must be italicized.

g) Page 2: “expression is required for both for the visually impaired and multisensory tangible interaction.”

h) Page 3: “on the morphed structure by photothermally induced in stiffness modulation at localized regions of the polymer bilayer.”

i) Page 5: “elongation exceeding 300%. At the same time, higher GNP content causes more softening of the GNPEs, accompanied by a weakening their ultimate tensile strength and ductility”

j) Page 7: “higher GNP contents in the PDMS matrix reduces of”

k) Page 22: “were established on a metal PCB with”, “For morphing actuation tests at thirty-six areas of the PtBA-GNPE bilayer, we prepared”

[Authors]: Thank you for important comments. As considering reviewer’s comments, we carefully reviewed and revised our manuscript including the discrepancies, which were pointed out by reviewer, with the assistance of professional English language editing service for scientific papers.

Reviewer #2 (Remarks to the Author):

This manuscript reports a photothermally-modulated height-renderable morphable tactile display which can tune both stiffness and height of the 3D Gaussian protrusion by introducing graphene nanoplatelet (GNP)-polydimethylsiloxane (PDMS) composite elastomer (GNPE)/photo-crosslinked poly(tert-butyl acrylate) (PtBA) bilayer. The two major properties were controlled by photothermal heat and pneumatic pressure, respectively. Although the materials are well known, there is still a lot of research being proceed and good application demonstration have been shown. Therefore, this manuscript has sufficient originality and novelty. Detailed analysis and experiment methods are easy to understand. I suggest some additional comments and questions below.

1. I think the authors could add more latest references to photothermal related papers in the literature for readers.

[Authors]: Thank you for kind comment. As faithfully considering reviewer’s comment, we additionally described the latest studies on organic or inorganic materials for photothermal heating, citing the references in the second paragraph of the chapter, “Materials for stiffness-tunable polymer and light-absorbing elastomer”. In our revised manuscript, the modified part including a list of references was

highlighted by using bright blue color.

2. The protruding deformation in the figures and videos seems large that it is difficult to touch the multiple protrusions with the index finger. Optionally, is there an easy way to reduce the size of the protruding deformation to similar to the size of Braille (1-2 mm)?

[Authors]: Thank you for important question. In this study, we have focused on demonstration of programmably height-renderable morphable tactile display responding to photothermal-pneumatic stimulus. Since we are also aware of the necessity for miniaturizing the active areas, we have a plan to reduce the size of the active areas, with increase in their spatial resolution. We believe that our morphable tactile display, in the near future, can produce the protruding deformation to similar to the size of Braille by constructing a hardware system arranging the smaller NIR-LEDs closely, with sophisticated design of the metallic pneumatic chamber.

3. In figure 5a (step II), it seems that the surface temperature of the bilayer can rise up to 125 oC. How long does it take to cool down to room temperature (can touch fingers)?

*[Authors]: Thank you for kind question. The surface temperature restored by 50% in 2.6 s and 94% in 10 s. Humans are known to feel heat pain when their skin temperature reaches ~45°C [ref. *]. Even if the maximum temperature at the center of the active area is 125 °C, the area where the surface temperature of the PtBA-GNPE bilayer is above 45 °C is very narrow, within 5 mm in diameter, as shown in Fig. 4e, and the total amount of thermal energy is quite small. Therefore, due to the narrow contact area and low thermal conductivity of PtBA ($<0.27 \text{ W}\cdot\text{m}^{-1}\cdot\text{K}^{-1}$, Supplementary Table 2), the skin temperature heats up slowly compared to the cooling rate of the bilayered film when the finger touches it. In our tests, no thermal pain was felt on the skin after 3 seconds of cooling.*

[] Darian-Smith, I. & Johnson, K. O. Thermal sensibility and thermoreceptors, J. Invest. Dermatol. 69, 146-153 (1977).*

4. It would be better to provide a height vs. temperature profile in the blank space in figure 3.

[Authors]: Thank you for kind comment. Since we intended to organize Fig. 3 with data plots presenting thermo-mechanical and thermo-electrical properties of the PtBA and GNPEs, the results related to actuation performance of our morphable tactile display, such as deformation height (D_{height}) with temperature, were placed to Fig. 5 (Please see the Fig. 5d).

5. The authors compared the thermal conductivity and elastic modulus of the materials enabling photothermal heating. How about comparing the response time and protrusion height (or size) with other displays? And it would be better to summarize these comparison values in a table in supplementary information.

[Authors]: Thank you kind suggestion. As faithfully considering reviewer's comment, we prepared a

supplementary table comparing deformation performance (including the response time and the maximum deformation height) of our tactile display with others, classifying the tactile displays according to driving force. We added the table with a list of references to the supplementary information as below:

Supplementary Table 3. Performance comparison of tactile displays.

Driving force	Activated layer	Response time for deformation (s)	Change in D_{height} (Steps)	Maximum D_{height} (mm)	Maximum D_{height} /Diameter (mm/mm)	Reference	
Pneumatic pressure	BS80-AA5	1	2	0.5-0.7	0.36-0.47	1-3	
	MM4520	2.5	3	0.3-0.4	0.13	4	
Joule heating	Vapor pressure	Silicone elastomer	13	2	1.3	0.43	5
	SMA* (TiNiCu) actuation	Spring structure	Not provided	>14 (estimated)	0.08	0.053	6
Electrical Stimulus	Fluidic pressure	Silicone elastomer	Not provided	>14 (Estimated)	0.05	0.1	7
			0.005	2	0.36-0.5	0.17	8
	Dielectric elastomer actuation	Not provided	14	0.471	0.24	9	
	IPMC* actuation	1	>14 (Estimated)	0.55	0.16	10	
Photothermal heating	Volume phase transition	PNIPAAm	Several seconds	2	0.25	0.83	11
	Pneumatic pressure	PtBA-GNPE	3	14	1.42	0.36	Our work

*SMA and IPMC are short for shape memory alloy and ionic polymer-metal composite, respectively.

6. Would you provide a photothermal conversion efficiency of graphene nanoplatelet (GNP)-polydimethylsiloxane (PDMS) composite elastomer?

[Authors]: Thank you for kind suggestion. As faithfully considering reviewer's comment, we calculated photothermal conversion efficiency of the GNPE 2.0 by applying its material properties with a temperature profile that was additionally achieved under NIR-LED light irradiation to the thermal equilibrium energy balance equation. As a result, we could estimate the photothermal conversion

efficiency of the GNPE to be 91.7%. In our revised manuscript (page 10), we additionally described that “the light absorbed in the GNPE is converted into thermal energy with a photothermal conversion efficiency of 91.7% (Supplementary Note 2)”. The modified part was highlighted by using bright blue color.

Furthermore, we added the incorporated references [1-4] to a list of supplementary references. A detailed procedure for estimating the photothermal conversion efficiency is also provided in Supplementary Note 4 as follows:

Supplementary Note 4. Estimation of the photothermal conversion efficiency of the GNPE

When light of wavelength λ is incident on a photothermal conversion material with power I_0 , the absorbed light energy within the material is converted into thermal energy with a photothermal conversion efficiency, denoted as η_{PCE} . This converted heat flow causes a temperature increase in the material over time and some of this thermal energy is dissipated by convection to the atmosphere, expressed as \dot{Q}_{loss} , and to the substrate \dot{Q}_{sub} . The energy balance equation at thermal equilibrium can therefore be derived as follows [1-4]:

$$I_0(1 - 10^{-\alpha(\lambda)}) \times \eta_{PCE} = \sum mc_p \frac{dT}{dt} + \dot{Q}_{loss} + \dot{Q}_{sub}$$

where $\alpha(\lambda)$ is the absorbance of the material at a given wavelength λ , and mc_p is the thermal capacitance of the material and substrate.

The convective heat loss \dot{Q}_{loss} is characterized as $hA(T - T_\infty)$, where h is the convective heat transfer coefficient, A is the surface area of the material for convection, T is a temperature of the material and T_∞ is the ambient temperature for the measurements. In a steady state, the temperature change with time ($\frac{dT}{dt}$) should be zero, and when a substrate with negligible photothermal conversion effect is used (*i.e.*, $\dot{Q}_{sub} = 0$), the photothermal conversion efficiency, η_{PCE} , can be simply expressed as follows:

$$\eta_{PCE} = \frac{hA(T - T_\infty)}{I_0(1 - 10^{-A(\lambda)})}$$

In this equation, the term hA is associated with the time constant (τ) of the specimen for the cooling process and is expressed as $\tau = \sum mc_p / hA$. This parameter can be determined experimentally by measuring the time taken for the temperature to fall from the steady-state temperature (T_{ss}) under light irradiation to $T_{ss} - 0.632(T_{ss} - T_\infty)$ during the cooling period.

To evaluate the photothermal conversion efficiency of GNPE, a 100 μm thick GNPE sample on PET substrate was cut into pieces, with an area of 36 mm \times 33 mm. The GNPE sample was irradiated with a I_0 of 16.7 mW/cm² at a λ of 940 nm using an LED (Luminus SST 10-IRD-B50-U940) positioned at a distance of 62.5 mm above the sample. The light output of the LED was quantified using a photodetector (Newport 918D-ST-SL) connected to a handheld photometer (Newport 1919-R).

Simultaneously, the temperature variation of the GNPE sample was recorded using a thermal camera (FLIR A6781 MWIR) at a frame rate of 50 frames per second, as shown in **Supplementary Fig. 10**. The material properties used to calculate the photothermal conversion efficiency are summarized in **Supplementary Table 1**. As a result, the photothermal conversion efficiency of GNPE is calculated to be 91.7%, which is comparable to those of carbon materials, which range from 83.73 to 93.4% [1-4].

Supplementary Fig. 10. Temperature profile of GNPE measured under LED irradiation.

Supplementary Table 1. Parameters used to calculate the photothermal conversion efficiency of GNPE.

Parameter	GNPE	PET
Density, ρ [$\text{g}\cdot\text{cm}^{-3}$]	1.054	1.37
Specific heat capacity, c_p [$\text{J}\cdot\text{g}^{-1}\cdot\text{K}^{-1}$]	1.415	1.0
Thermal capacitance, mc_p [$\text{mJ}\cdot\text{K}^{-1}$]	33.556	30.825
Absorbance, $\alpha(940\text{ nm})$	1.4745	-

Steady-state temperature, T_{ss} : 31.03 °C and ambient temperature, T_{∞} : 22.25 °C
Time constant, τ : 16.4 s

Incorporated references

1. Ren, H. *et al.* Hierarchical graphene foam for efficient omnidirectional solar–thermal energy conversion. *Adv. Mater.* **29**, 1702590 (2017).

2. Yang, H. *et al.* Developed carbon nanotubes/gutta percha nanocomposite films with high stretchability and photo-thermal conversion efficiency. *J. Mater. Res. Technol.* **9**, 8884-8895 (2020).
3. Luo, W. *et al.* Efficient enhancement of photothermal conversion of polymer-coated phase change materials based on reduced graphene oxide and polyethylene glycol. *J. Energy Storage* **78**, 109950 (2024).
4. Lv, S. *et al.* Flexible highly thermally conductive biphasic composite films for multifunctional solar/electro-thermal conversion energy storage and thermal management. *J. Clean. Prod.* **426**, 139004 (2023).

REVIEWERS' COMMENTS

Reviewer #1 (Remarks to the Author):

The provided rebuttal and the revision is satisfactory. The authors have addressed the comments to the previous review satisfactorily. I recommend the article for publication subject to the following minor corrections.

Minor corrections:

This statement in the introduction says "recently", "Recently, height-adjustable tactile displays have been introduced in the form of tactile pin arrays that can express 2.5D images for tangible interaction". But the referred articles were published 6-26 years ago.

"and high-dimensional information of objects" should read "and for high-dimensional information of objects".

Please revise this "comparable to the force of pin-type tactile display that was determined to conduct normal haptic exploration for visually-impaired people".

"achieved during 50 continuative actuation cycles"- "continuative" should read "continuous".

Reviewer #2 (Remarks to the Author):

The authors took into account all reviewer's remarks and comments. This paper can be published without further changes.